# Electric Heating Behavior of Reduced Oxide Graphene/Carbon Nanotube/Natural Rubber Composites with Macro-Porous Structure and Segregated Filler Network

**DOI:** 10.3390/polym12102411

**Published:** 2020-10-19

**Authors:** Yanhu Zhan, Yuchao Li, Yanyan Meng, Qian Xie, Marino Lavorgna

**Affiliations:** 1School of Materials Science and Engineering, Liaocheng University, Liaocheng 252000, China; liyuchao@lcu.edu.cn (Y.L.); mengyanyan@lcu.edu.cn (Y.M.); xieqian@lcu.edu.cn (Q.X.); 2Institute of Polymers, Composites and Biomaterials, National Research Council, P.le Fermi, 1-80055 Portici, Naples, Italy

**Keywords:** electric heater, reduced graphene oxide, carbon nanotube, segregated network, macro-porous structure

## Abstract

Conductive polymer composites with carbonaceous fillers are very attractive and play a significant role in the field of electric heaters owing to their lightweight, corrosion resistance, and easy processing as well as low manufacturing cost. In this study, lightweight reduced oxide graphene/carbon nanotube/natural rubber (rGO/CNT/NR) composites were fabricated by a facile and cost-effective approach, which consists of rGO assembling on rubber latex particles and hydrogels formation due to the interaction network established between carbonaceous fillers and subsequent mild-drying of the resulting hydrogels. Thanks to the amphiphilic nature of GO sheets, which can serve as a surfactant, the hydrophobic CNTs were easily dispersed into water under ultrasound. On the basis of both the high stable rGO and CNTs suspension and the assembling of rGO on rubber latex, a three-dimensional segregated network of CNT and rGO were easily constructed in macro-porous composites. Either the segregated network and macro-porous structure endowed the resulting composites with low density (0.45 g cm^−3^), high electrical conductivity (0.60 S m^−1^), and excellent electric heating behavior, when the weight content of rGO and CNTs are 0.5% and 2.5%, respectively. For electric heating behavior, the steady-state temperature of the above composites reaches 69.1 °C at an input voltage of 15 V.

## 1. Introduction

Electrical heaters, which are a kind of electrical resistor that converts electrical energy into thermal energy, have attracted increasing attention due to their widespread applications in many civil, industrial, and military fields as wearable devices, defogging and defrosting, and personal thermal management [1,2,3,4,5]. Based on the Joule’s first law ($Q=Pt=V^{2}t/R$, where *Q*, *P*, *V*, *R,* and *t* are the generated heat, the electric power, the applied voltage, the resistance of the electrode, and the heating time, respectively), the generated heat is inversely proportional to the resistance of an electrode at a fixed applied voltage [6,7]. Clearly, a lower electrical resistance leads to a higher generated heat. Therefore, metallic materials have been commonly used as electrical heaters owing to their low resistance in past years. Metallic materials, however, cannot meet the requirements and open for new perspective applications because of their features, such as stiffness, easy-corrosion, high density, and so on. Recently, the rise of conductive polymer composites (CPCs), possessing lightweight, corrosion resistance, easy processing, and low manufacturing cost, limits the monopoly of metallic materials and CPCs, which, consequently, become the best candidate for the preparation of electrical heaters [8,9].

Generally, the electrical conductivity of CPCs depends on the formation of a continuous path of conductive filler in an insulating polymer matrix. It is well known that conductive path is related with types and distribution of fillers. The conductive fillers with a high aspect ratio, including carbon based fillers (i.e., carbon nanotube (CNT) [4,10], carbon fiber [11], graphene, or reduced graphene oxide (rGO) [4,12,13,14]), MXene [15,16], and silver nanowires [1,17], are commonly utilized in order to construct conductive path with low filler loadings. As compared with MXene, which easily decompose at air and silver nanowires that are significantly expensive, carbon-based fillers meeting the requirement of practical applications owing to their thermal stability and relatively low-cost. In addition, it’s well documented that the spatial distribution of the nanoparticles within the polymer matrix, which can be classified as random distribution, layered assembly, and segregated architectures, affects the formation of a conductive path [18]. Among the above-mentioned three morphological organizations, the segregated filler network wherein the conductive filler is arranged to produce a three-dimensional network, which percolates the polymer volume [18], is particularly attractive. In fact, alongside with enhanced mechanical properties, this morphology allows a high electrical conductivity with the lowest percolation threshold, as compared with the other morphological organization [19,20,21]. Therefore, the past decade has witnessed an almost explosive growth of CPCs with a segregated carbon-based fillers network [10,13,22,23,24]. Yu et al. [25] first demonstrated that the electrical conductivity of CNT/poly(vinyl acetate) increased dramatically after incorporating CNTs in the segregated network. Zhan et al. [4] reported that the rGO/natural rubber (NR) composites in which the rGO nanosheets located at the rubber particles interface to construct the segregated network exhibited good electric heating behavior. The steady-state temperature of the composites reached the value of 152 °C at the applied voltage of 20 V. The anisotropic layer-by-layer CNT/boron nitride/NR composites with segregated network, designed by Zhan and his colleagues, showed an ultra-high surface temperature (170 °C) in 30 s under an electric potential of 7.5 V [10]. Pu et al. [26] found that a segregated CNTs network endowed CNT/polyurethane composites with excellent electric heating behavior, consequently, leading to an extraordinary self-healing property.

Moreover, lightweight porous CPCs characterized by low density, high specific mechanical properties, and high capability to adsorb impact-energy have great potential in several application sectors [27,28]. The electrical conductivity of CPCs in which the conductive fillers were randomly dispersed improved significantly after a foaming procedure because of the redistribution of conductive fillers and their constraint in a reduced volume (struts and walls of the porous material) with consequent reduction of a distance between adjacent conductive fillers [29,30]. For example, the electrical percolation threshold of CNT/polypropylene foams decreased from 0.50 vol% to 0.09 vol% after foaming [30]. Liu et al. [29] also found that the conductivity of the foamed sample increased, resulting in a decrease of the percolation threshold from 9 to 6 wt %. On the contrary, the electrical conductivity of CPCs with segregated filler network decreases after foaming because of the formation of a discontinuous segregated network, even at low foaming degrees. Some authors of this work noted that the conductivity of CNT/NR composites with a segregated network were 0.26 S m^−1^ and 4.90 × 10^−4^ S m^−1^ before and after the foaming process, respectively [31]. In order to address the decreasing electrical conductivity of composites with a segregated network, Wang et al. [32] first prepared porous polylactic acid beads by supercritical CO_2_ and then wrapped CNT upon the porous beads in order to construct a segregated CNT network after the steam-chest molding process. The electrical conductivity of the obtained foams was as high as 6.3 S m^−1^ at an extremely low filler concentration of 0.0054 vol%. [32] Unfortunately, it is very difficult to obtain porous rubber beads by the supercritical CO_2_ foaming procedure due to their low glass transition temperature. Therefore, it is a big challenge to fabricate conductive rubber foams with an effective segregated network and high electrical conductivity simultaneously.

Herein, rGO/CNT/NR (GCR) macro-porous materials with conductive fillers exhibiting the three-dimensional segregated morphology were fabricated by investigating a multi-step approach, which consists of the following sequential steps: (a) rGO and CNTs latex assembling during in-situ GO reduction in the presence of NR latex particles, (b) rGO/NR hydrogel formation and, finally, (c) hydrogel drying to form composites with macro-porosity. SEM and TEM images confirmed the coexistence of segregated network and porous structure. In addition, the mechanical, electric heating and thermal performance of rGO/CNT/NR macro-porous composites were carefully examined.

## 2. Materials and Methods

### 2.1. Material

Natural rubber latex (NRL, solid content: 60 wt %) was supplied by Zhengmao Petrochemical Co., Ltd. (Maoming, China). Carbon nantubes (NC 7000, diameter: 10 nm, length: 1.5 μm, density: 1.75 g cm^−3^, electrical conductivity: 10^6^ S m^−1^) were purchased from Nanocyl S.A., Sambreville, Belgium. Flake graphite (~75 mm) was obtained from Qingdao Tianhe Graphite Co., Ltd. (Qingdao, China). Potassium permanganate (KMnO_4_), potassium hydrate, hydrazine hydrate, sulfuric acid (H_2_SO_4_), and hydrochloric acid (HCl) were all of an analytical-grade and obtained from Sinopharm Chemical Reagent Co., Ltd. (Shanghai, China). Hydrogen peroxide (30%) was provided by Tianjin Zhiyuan Chemical Reagent Co., Ltd. (Tianjin, China). All agents are used without further purification.

### 2.2. Preparation of Macro-Porous rGO/CNT/NR Composites with a Segregated Network

GO was prepared by adopting a modified Hummer and Offeman method as described in previous papers [13,22,33,34]. Defined masses of GO and CNTs, as illustrated in Table 1, were dispersed into H_2_O (100 g) and ultrasonicated for 30 min. Several mass quantities of NRL were added into the above suspension containing GO and CNTs by stirring for 30 min. Then, hydrazine hydrate was rapidly poured into the GO/CNT/NR latex dispersions. The GO in the GO/CNT/NRL latex dispersion was reduced to rGO by hydrazine hydrate at 90 °C for 2 h. During the reduction, the aqueous dispersions turned to rGO/CNT/NR hydrogel, due to the formation of filler-filler interactions, which contribute to gelificate the structure. After drying at 70 °C for 6 h, the water was removed from the hydrogels and rGO/CNT/NR macro-porous composites were obtained. The resulting composites were denoted as G*_x_*C*_y_*R, where *x* and *y* are the rGO and CNTs concentration expressed as a weight percentage with respect to natural rubber, respectively. Significantly, with the procedure, it is not possible to get the porous CNT/NR materials.

### 2.3. Characterization

Field emission scanning electron microscope (FE-SEM) observations were conducted by using a Zeiss Ultra 55 apparatus (Zeiss, Oberkochen, Germany) at an accelerating voltage of 10 KV. In detail, the specimens were cut by a sharp knife and sputter-coated with gold before FE-SEM observation.

TEM was performed using a FEI TecnaiG^2^ F20 S-TWIN transmission electron microscope(FEI, Portland, OR, USA), operating at an accelerating voltage of 200 kV. GCR porous samples were first filled by epoxy resin and then cryomicrotomed by using a Leica EM UC6 equipment. The ultrathin cryosections of 70–80 nm thickness were collected and directly supported on a copper grid for observations of composite morphology.

The apparent density of porous composites was determined as the ratio of sample weight to its volume in accordance with the ISO 845 standard. The mass was measured by an analytical balance with a precision of 0.001 g whereas the sample volume was determined by the geometrical shape measuring the dimensions by using a micrometer with an accuracy of 0.001 mm. Five specimens for each composite were tested and the results were expressed as an average of the several measurements.

A DSC25 Differential Scanning Calorimetry (DSC, TA Instruments, New Castle, DE, United States) instrument was used to investigate the thermal properties of GCR macro-porous composites in a nitrogen-filled atmosphere. About 5~10 mg samples were cooled to −90 °C and then heated to 0 °C at a rate of 10 °C min^−1^.

Thermogravimetric Analysis (TGA) was carried out using a TGA 4000 (PerkinElmer, Groningen Netherlands) in a nitrogen-filled and air-filled atmosphere to characterize the thermal stability of samples. The samples were heated from 30 to 750 °C at a heating rate of 10 °C min^−1^.

Mechanical characterization was performed by using a universal testing machine (MTS E44) and submitting out the samples to compression tests with a deformation rate of 5 mm min^−1^.

Electrical conductivity was measured with a Keithley 2000 SourceMeter by connecting the electrodes to samples coated with silver paste to eliminate the effect of contact resistance. Five samples were measured for each composite and the result is expressed as an average of several values. To evaluate the electric heating behaviour, the silver-coated samples (10 × 5 × 2 mm^3^) were connected by copper wires to the DC power supply. An IR camera (FLIR one pro) was used to record the dynamic change of the surface sample temperature at different input voltages.

## 3. Results and Discussion

### 3.1. Structure and Morphology of Macro-Porous GCR Composites

The morphology and chemical structure of GO were described in our previous works [13,22,33,34,35]. The size of rGO as well as results from XRD, XPS, FTIR, and Raman characterizations were detailed in our previous paper [33]. Thanks to the amphiphilic nature of GO sheets, which possess polar groups, such as hydroxyl, carboxyl, and epoxy groups, and serve as surfactant, the hydrophobic CNTs can be easily dispersed into water under an ultrasound, as shown in Figure 1a. This is attributed to the fact that single CNT can be easily adsorbed onto the GO surface through π-π attractions between GO and CNTs, which, in turn, contribute to minimize their sedimentation in water dispersions [36]. Therefore, in this work, other surfactants, i.e., cetyltrimethylammonium bromide, 3-aminopropylmethyldimethoxysilane (HK550), and octylphenol polyoxyethylene ether (OP-10), are not added during the dispersing process of CNTs. Notably, the rGO/CNTs are relatively stable in water over a period of 24 h. After mixing with ivory NRL, the black GO/CNTs dispersion changes to a dark-gray color (Figure 1b). During the reduction process of GO, CNTs, and the generated rGO coated the surface of NR particles, and, as a result of the electrostatic and π-π interactions establishing between the latex particles, the water dispersion turns to the rGO/CNT/NR hydrogels. Significantly, the fillers coated on the surface of NR particles give rise to the formation of the segregated network, as shown in several examples as rGO/NR, [13] Fe_3_O_4_@rGO/NR, [22] CNT/NR, [10] and CNT/NR foam [31] reported by some of authors. In fact, the main reason for the formation of rGO/CNT/NR hydrogel is ascribed to the rGO and CNTs coated NR particles, which, firmly interconnecting to each other through strong interactions, preserve the porous structure against the forces generated by the nitrogen gas evolving during a chemical reduction. In other words, the entanglement effect of flexible rubber chains at the interface between two rubber particles results in the formation of hydrogels. The macro-porous GCR composites with the segregated filler network are obtained after removing water at 70 °C in the oven, as shown in Figure 1c. SEM and TEM were employed to characterize the structure and morphology of macro-porous GCR composites. Regular pores with a diameter of about 0.9 mm are homogeneously distributed throughout the sample volume, realizing a highly interconnected open structure, as shown in Figure 2a. The pores are separated by thick skeleton walls of about 0.2-mm thickness, which contribute to improve the mechanical property and electrical conductivity of the resulting composites [37,38]. In the inner walls, it is possible to observe the construction of an rGO and CNT segregated network (see Figure 2b). In addition, it is noted that the rGO nanosheets and CNTs coat the surface of porous cells, as displayed in Figure 2c,d. TEM is one of the most effective techniques to observe the segregated filler network [39]. Figure 3a,b shows clearly the segregated network of rGO and CNTs in the bulk material constituting the walls of porous cells. Importantly, the CNTs aggregates are not found, thus confirming the effectiveness of GO in enhancing the dispersion of CNTs in NR latex. The cross section of NR particles is roughly circular and the average diameter of particles is about 500 nm. Finally, the rGO layers and CNTs coated on the surface of the rubber latex particles, which are evenly distributed in the rubber matrix, form a continuous conductive network that enables us to enhance the electrical conductivity of the resulting porous composites [18].

### 3.2. Density of Macro-Porous GCR Composites

Density is a critical factor to evaluate the lightweight composites [40]. Figure 4a shows the density of GCR composites with different CNTs and rGO contents. It depends significantly on the CNTs and rGO content. In fact, when the ratio of CNT and rGO is fixed as 1:1, the density of GCR composites first sharply decreases and then slightly increases with the filler content. For example, the density of G_1.5_C_1.5_R composites is equal to 0.32 g cm^−3^, while that of G_0.5_C_0.5_R composites is 0.56 g cm^−3^. This is ascribed to the effect of the strong interface forces between rubber chains and fillers. In fact, when the filler content increases, the presence of strong interactions between the polymer and filler restricts the macromolecules movement and, thus, the porous structure resists the volume shrinkage during the drying process, leading to a low density. It is found that the density of G_0.5_C_2.5_R composites is equal to the value of 0.45 g cm^−3^, whereas the density of G_2.5_C_0.5_R is significantly lower and equal to 0.33 g cm^−3^. These results confirm that, for the macro-porous composites with the same filler content (3 wt %), the higher the CNTs content is, the higher the composite density is, suggesting the important role of rGO in resisting the shrinkage of composites’ volume. Notably, the low density is interesting in practical application to save materials and cost. The expansion ratio, which is an important parameter for porous materials, is the ratio of sample density before and after foaming [41,42] and is shown in Figure 4a. It is well known that the expansion ratio displays a contrary tendency with respect to the density. The porosity (*f*) of the fabricated composites is calculated from the density of GCR composites (*ρ_f_*) and the density of solid rGO/CNT/NR composites (*ρ_s_*) using Equation (1) [43].
(1)f=1−ρfρs

The porosity of GCR composites is shown in Figure 4b. G_1.5_C_1.5_R composites display the highest porosity (65.8%), while G_0.5_C_0.5_R composites possess the lowest one (41.2%).

### 3.3. Electrical Conductivity of Macro-Porous GCR Composites

Segregated morphology has been proven to be very effective, as compared with the random and layered morphology, in enhancing the electrical conductivity of composite materials [44]. In particular, for composites exhibiting a segregated filler network, the conductive fillers are located at the interfaces between the polymer particles and build-up a three-dimensional percolative path throughout the bulk material, leading to a low percolation threshold [4]. Electrical conductivity of macro-porous GCR composites is shown in Figure 4c. When the ratio of CNTs and rGO is 1:1, the electrical conductivity of GCR composites sharply increases by increasing the total content of CNTs and rGO. This is ascribed to the building-up of a perfect and effective conductive network by increasing the conductive filler content. For example, the electrical conductivity of G_0.5_C_0.5_R, G_1.5_C_1.5_R, and G_3_C_3_R are 4.7 × 10^−4^, 0.093, and 0.19 S m^−1^, respectively. In the case of composites with the same total content (3 wt %) of carbonaceous filler (i.e., the mixture of CNTs and rGO), the electrical conductivity is strongly dependent on the CNTs content. The higher the CNTs content is, the higher the electrical conductivity is. For instance, the electrical conductivity of macro-porous G_0.5_C_2.5_R composites is equal to 0.60 S m^−1^, while that of G_2.5_C_0.5_R macro-porous composites is only 0.081 S m^−1^. The result may be attributed to the fact that the electrical conductivity of CNTs (10^6^ S m^−1^) is higher than that of rGO (2420 S m^−1^) [45]. The specific electrical conductivity of macro-porous GCR composites are reported in Figure 4d. It is found that the electrical conductivity of GCR composites is not dependent on their density. Therefore, the excellent electrical conductivity obtained for the porous composites, thanks to the formation of the segregated filler network, confirms that the materials can be successfully used in the field of electric heating systems.

### 3.4. Electric Heating Behavior of Macro-Porous GCR Composites

Electrical energy dissipated by the Joule effect was monitored by IR imaging of the samples’ surface after inputting a voltage [4,10]. The samples’ surface exhibits a somewhat uniform temperature distribution (Figure 5a) when input voltages of more than 7 V are applied. The time-dependent temperature profiles of G_3_C_3_R composites are shown in Figure 5b. When the voltage is applied to G_3_C_3_R composites, the surface temperature gradually increases and then levels off to a steady state. The surface temperature of macro-porous G_3_C_3_R composites increased from 23.2 °C to 67.0 °C in 30 s when the applied voltage changed from 7 V to 20 V, suggesting a sharp non-linearity (switching effect) in the heat-voltage [46]. These results confirm that the macro-porous G_3_C_3_R composites can be used as a heat dissipater by applying heat-voltages similar to those previously applied for graphene/NR, i.e., 10 V [4]. An input voltage of 15 V was applied to the macro-porous GCR composites with good electrical conductivity for investigating the effect of the ratio of CNTs and rGO, and filler contents on their electric heating behavior. The results are shown in Figure 5c. G_0.5_C_0.5_R composites were not tested as electric heater materials because of their low electrical conductivity (4.7 × 10^−4^ S m^−1^, Figure 4c). The stationary temperature of all samples is reached rapidly. The steady-state temperature of composites with the same total filler content changes from 30.7 °C to 69.1 °C when the ratio of CNTs and rGO increases from 1/5 to 5/1, which is attributed to the increment of the electrical conductivity of GCR with the increase of CNTs content, as shown in Figure 4c. The surface temperature of G_3_C_3_R composites (56.9 °C) is lower than that of G_0.5_C_2.5_R composites with electrical conductivity of 0.60 S m^−1^ (69.1 °C), resulting from the lower electrical conductivity of G_3_C_3_R composites (0.19 S m^−1^).

To pursue a more detailed analysis of the characteristic thermal properties of macro-porous composite electric heater materials, two parameters, namely the characteristic time growth (*τ_g_*) and the efficiency of heat transfer (*h_r+c_*), were measured. Analyzing the curves in Figure 5c, the time-dependent temperature curves can be divided into two stages: the temperature growth stage (heating) and the equilibrium stage (i.e., heat gain by working power is equal to heat loss by radiation and convection). In the first stage, the *τ_g_* during applied power can be calculated by an exponential growth function, as shown below [8,11,47].
(2)$$\left(\frac{T_{t}-T_{0}}{T_{m}-T_{0}}\right)=1-e^{-t/\tau_{g}}$$
where *T_0_* and *T_m_* are the initial and maximum temperature, respectively, *T_t_* represents the temperature at time *t*. For all composites, the *τ_g_* values can be obtained by fitting the data in the first stage (5~40 s) of temperature-time curves in Figure 5c, and the results are illustrated in Table 2. In the case of GCR porous composites with the total content of 3 wt %, the values of *τ_g_* decrease sharply by increasing the ratio of CNTs to rGO. This demonstrates that GCR composites with higher CNTs content exhibit a faster temperature response to applied voltage [8]. The *τ_g_* value of G_3_C_3_R composites is higher than that of G_0.5_C_2.5_R, resulting from their lower electrical conductivity (as shown in Figure 4c).

In the second stage, the heat transferred by radiation and convection, *h_r+c_*, is expressed by: [8]
(3)$$h_{r+c}=\frac{I_{c}V}{T_{m}-T_{0}}$$
where *I_c_* is the steady state current. The values of *h_r+c_* are summarized in Table 2. The lower *h_r+c_* value indicates the lower efficiency of heat transfer by effective radiation and convection, suggesting high electric power efficiency by consuming a relatively low electrical energy to maintain a steady state of maximum temperatures at given applied voltages [47].

### 3.5. Compression Stress of Macro-Porous GCR Composites

Figure 6 shows the effect of the filler contents on the compression stress-strain curves of macro-porous GCR composites with a segregated network. For the macro-porous GCR composites with the same ratio of CNTs and rGO, the compression stress significantly increases from 0.11 MPa to 0.30 MPa when the filler content increases from 1 wt % to 6 wt %. It is attributed to the segregated filler network, which works as a structural scaffold inside the composites, to the extraordinary modulus of fillers, and the strong interface force between the filler and rubber matrix [33]. Furthermore, the compression stress of macro-porous GCR composites with filler contents equal to 3 wt % depends on the ratio of CNTs to rGO. The stress of macro-porous G_0.5_C_2.5_R composites is 1.5 times higher than that of macro-porous G_2.5_C_0.5_R composites, implying that the porous materials resist to higher external stress. The higher expansion ratio (see Figure 4a) is responsible for the lower stress.

### 3.6. Thermal Stability of Macro-Porous GCR Composites

The thermal stability of electric heating materials affects the temperature range wherein the samples can operate. Figure 7 shows the thermal degradation curve of macro-porous GCR composites with different fillers content in a nitrogen-filled (Figure 7a) and air-filled (Figure 7b) atmosphere. For the thermal degradation curves of GCR composites in the nitrogen atmosphere (Figure 7a), the onset degradation temperature of G_0.5_C_0.5_R composite is the lowest among all the samples. Macro-porous GCR composites with higher filler contents exhibit higher onset degradation temperature, ascribing to the presence of carbon-based fillers that contribute to thermally protect the polymer and hinder the transport of degradation gases/volatiles [48]. The thermal stability of macro-porous GCR composites were measured in the air atmosphere in order to evaluate their safe use as electrical heating materials, under ambient conditions, and the results are shown in Figure 7b. The inset image of Figure 7b displays that CNTs possess the most excellent thermal stability as compared with rGO and NR without a macro-porous structure. In the case of macro-porous GCR composites, their onset degradation temperature exceeds the 250 °C, which is the onset for the pristine NR without a macro-porous structure. Therefore, the introduction of fillers and a porous structure contribute to increase the thermal stability of NR. Based on Figure 7, macro-porous GCR composites can be safely used below 200 °C, which represents the maximum temperature reached during the electrical heating experiments.

### 3.7. Glass-Transition Temperature of Macro-Porous GCR Composites

The glass transition temperature (*Tg*) is the lower limit temperature of rubber composites. Therefore, DCS curves of macro-porous GCR composites at different fillers concentration were measured and shown in Figure 8. All samples were cooled and kept at −90 °C for 10 min and then heated to 0 °C. An apparent glass transition region can be observed in curves of all samples. The curves highlight that the addition of rGO and CNTs do not affect the *Tg* (around −62 °C). Similar results are obtained for rGO/NR composites [33] and CNT/NR foams obtained by a supercritical CO_2_ foaming procedure [31]. The low *Tg* implies that macro-porous GCR materials can operate at the ultra-low temperature.

## 4. Conclusions

rGO/CNT/NR porous materials with segregated morphology were realized through a simple approach consisting of in-situ GO reduction in the presence of CNTs. Thanks to excellent hydrophilic property of GO, used as a surfactant, CNTs can be uniformly dispersed into water. During the reduction procedure of GO nanosheets, rGO nanosheets, together with CNTs, coated the surfaces of NR latex particles, contributing to the formation of the segregated filler morphology, which endowed high electrical conductivity in the obtained porous composite materials. The developed materials exhibited excellent electric heating behavior. The steady-state temperature of the macro-porous composites containing 0.5 wt % rGO and 2.5 wt % CNTs, characterized by a low density (0.45 g cm^−3^), reached the value of 69.1 °C with an input voltage of 15 V, whereas their electrical conductivity was 0.60 S m^−1^. Moreover, the stress of macro-porous G_0.5_C_2.5_R composites possessed a high stress (0.21 MPa) at the compression strain of 30%. All results confirmed that the macro-porous rGO/CNT/NR composites had great potential as functional materials for electric heaters. The method referred to in this manuscript can open a window in preparing the rubber composites with a macro-porous structure and segregated network.

## Figures and Tables

**Figure 1 polymers-12-02411-f001:**
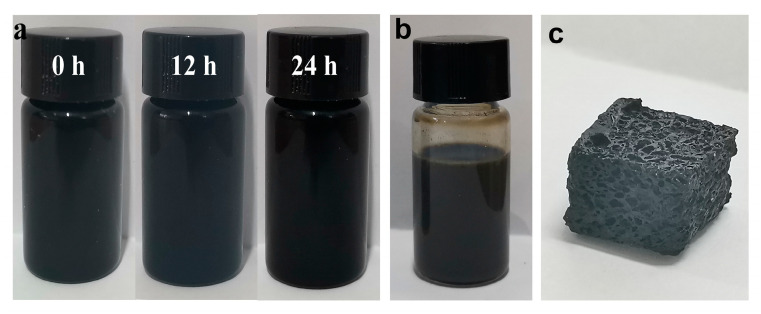
The photo images of GO/CNT suspension standing for different amounts of time (**a**), GO/CNT/NR latex (**b**), and GCR macro-porous composites (**c**).

**Figure 2 polymers-12-02411-f002:**
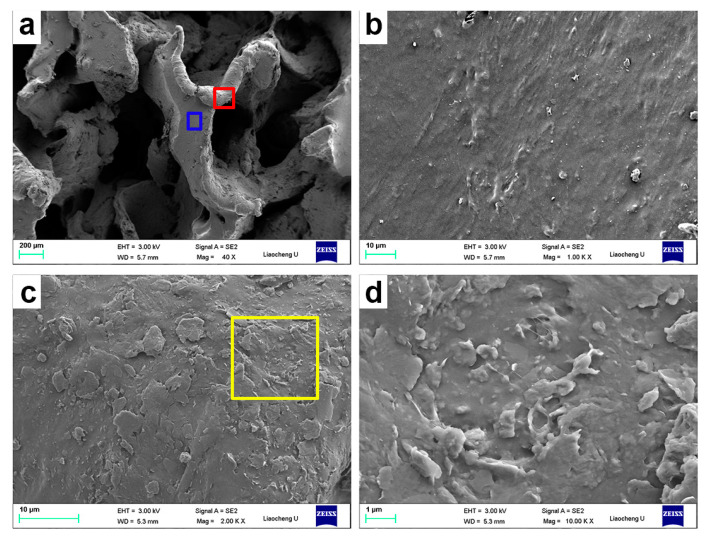
SEM images of G_3_C_3_R macro-porous composite (**a**). Images (**b**) and (**c**) are the magnification of blue and red zones of image (**a**), respectively. Image (**d**) is the magnification of the yellow zone of the image (**c**).

**Figure 3 polymers-12-02411-f003:**
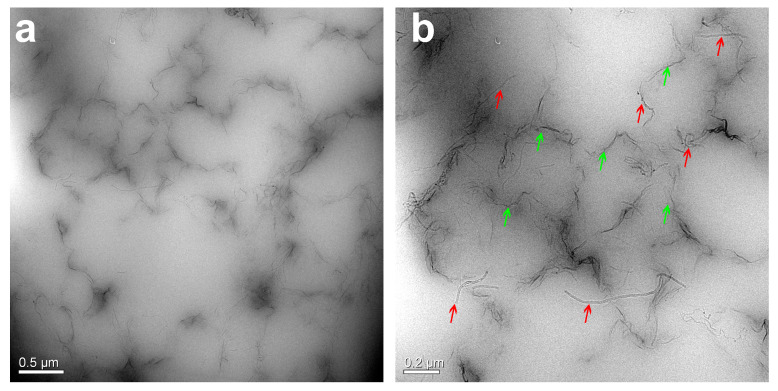
TEM images of G_3_C_3_R macro-porous composite (**a**). Image (**b**) is the magnification of image (**a**). Red and green arrows represent CNTs and rGO sheets, respectively.

**Figure 4 polymers-12-02411-f004:**
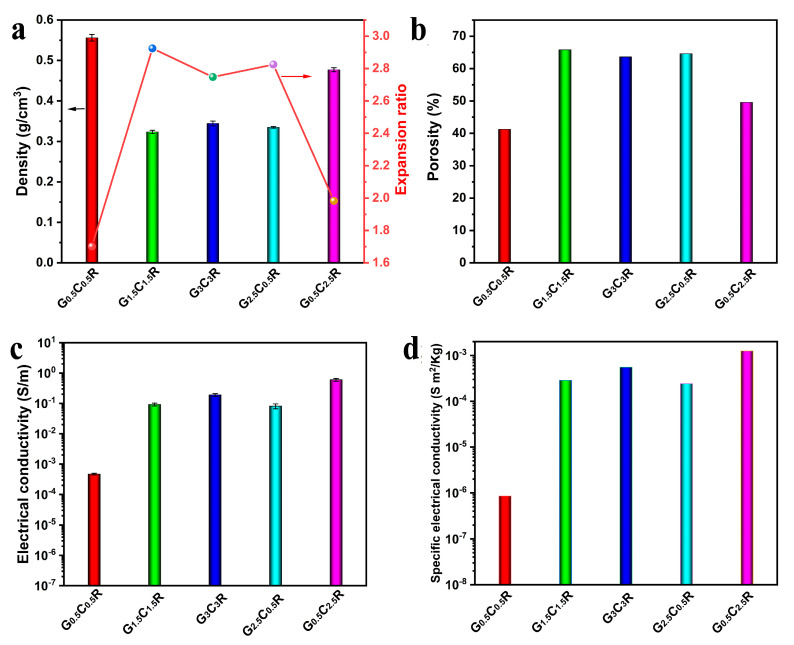
Density, expansion ratio (**a**), porosity (**b**), electrical conductivity (**c**), and specific electrical conductivity (**d**) of porous GCR materials.

**Figure 5 polymers-12-02411-f005:**
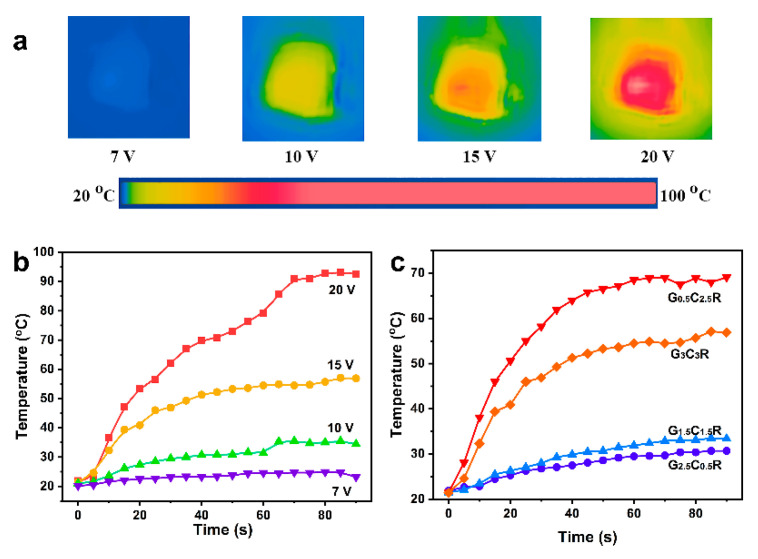
IR image of the G_3_C_3_R composite surface after inputting different voltages for 90 s (**a**). The time-dependent temperature profiles of G_3_C_3_R composites for the different input voltages (**b**). The time-dependent temperature profiles of GxCyR composites with an input voltage of 15 V (**c**).

**Figure 6 polymers-12-02411-f006:**
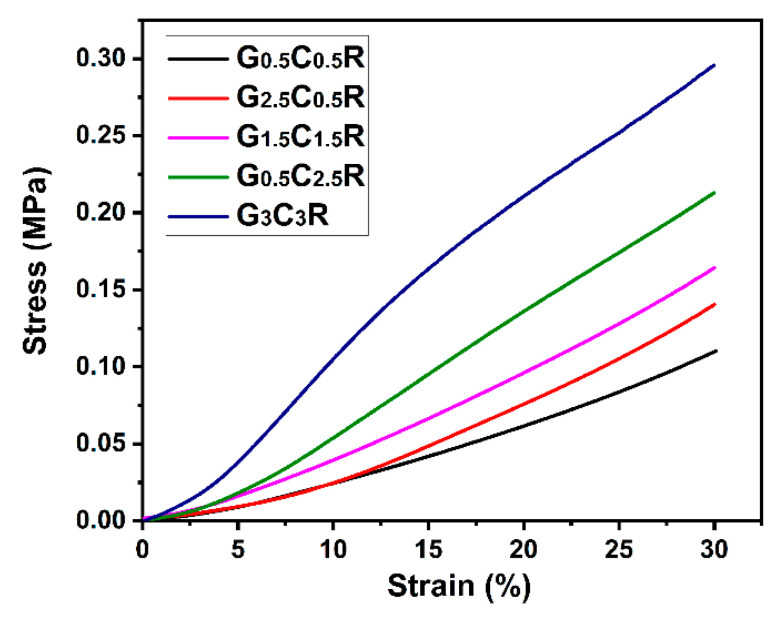
Compression stress-strain curves of GCR macro-porous composites.

**Figure 7 polymers-12-02411-f007:**
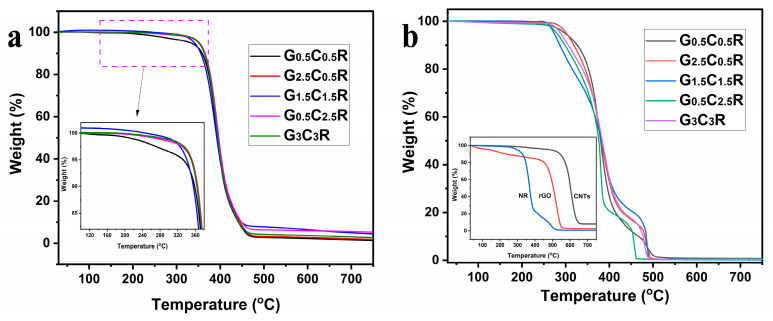
Thermal degradation curves of macro-porous GCR composites at different fillers concentration in a nitrogen-filled (**a**) and air-filled (**b**) atmosphere. The inset image of Figure 7b is the TGA curves of CNTs, rGO, and pure NR in the air-filled atmosphere.

**Figure 8 polymers-12-02411-f008:**
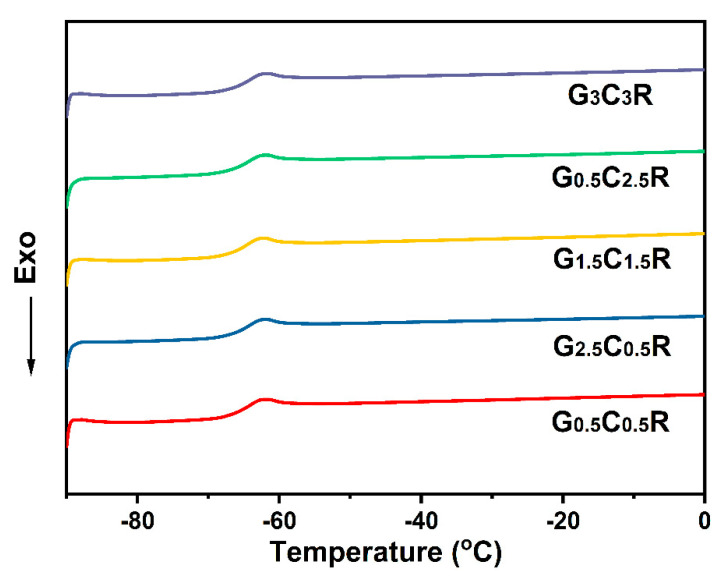
DSC curves of macro-porous GCR composites at different fillers concentration.

**Table 1 polymers-12-02411-t001:** Formulations of GCR marco-porous composites.

Sample	GO (g)	rGO (g)	CNT (g)	H_2_O (g)	NRL (60%)	N_2_H_4_(mL)
G_0.5_C_0.5_R	0.25	0.15	0.15	100	50	0.25
G_0.5_C_2.5_R	0.08	0.05	0.25	100	16.67	0.08
G_1.5_C_1.5_R	0.25	0.15	0.15	100	16.67	0.25
G_2.5_C_0.5_R	0.42	0.25	0.05	100	16.67	0.42
G_3_C_3_R	0.5	0.30	0.30	200	16.67	0.50

**Table 2 polymers-12-02411-t002:** Characteristic parameters (*τ_g_* and *h_r+c_*) for electric heating behavior of GCR porous composites.

Sample	Voltage (V)	*τ_g_* (s)	*h_r+c_* (mW/°C)
G_2.5_C_0.5_R	15	35.0	3.4
G_1.5_C_1.5_R	15	32.6	3.0
G_0.5_C_2.5_R	15	20.4	2.4
G_3_C_3_R	15	23.3	1.5

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
