# Peer review of "Electric Heating Behavior of Reduced Oxide Graphene/Carbon Nanotube/Natural Rubber Composites with Macro-Porous Structure and Segregated Filler Network"

_polymers, 2020, doi:10.3390/polym12102411_

Round 1

Reviewer 1 Report

In the article “Electric Heating Behavior of Reduced Oxide Graphene/Carbon Nanotube/Natural Rubber Composites with Macro-Porous Structure and Segregated Filler Network” authors talk about the use of graphene/CNT composites as conductive polymers and application in electric heaters. I recommend the publication of this article after the authors address the questions I mention below.

  1. Given the promising conductive nature of the composite materials, I am curious to know if these results are replicable? Did the authors have multiple measurements for density and conductivity of the composites? 
  2. Can the authors quantify the porosity of the composites? Is there a relation between porosity and conductivity?
  3. Can any other material be used instead of NR Latex? If yes, what are the alternatives and why?

Author Response

Reviewer 1:

Q1. Given the promising conductive nature of the composite materials, I am curious to know if these results are replicable? Did the authors have multiple measurements for density and conductivity of the composites? 

A1. Authors thank the reviewer for this comment. In order to get accurate and reliable density and electrical conductivity values for the GCR composites, five specimens were tested for every case and the results are reported as average of five determinations. The error bars of the density and conductivity for the GCR composites have been added in the revised manuscript.

Figure 4 with error bars and the text added to the manuscript is reported hereafter:

Figure 4. Density, expansion ratio (a), porosity (b) electrical conductivity (c) and specific electrical conductivity (d) of porous GCR materials

The apparent density of porous composites was determined as the ratio of sample weight to its volume in accordance with ISO 845 standard. ….. Five specimens for each composite were tested and the results were expressed as average of the several measurements.” (Line 135-139, Page 3)

“Electrical conductivity was measured with a Keithley 2000 SourceMeter by connecting the electrodes to samples coated with silver paste to eliminate the effect of contact resistance. Five samples were measured for each composite and the result is expressed as average of the several values.” (Line 149, Page 4)

Q2. Can the authors quantify the porosity of the composites? Is there a relation between porosity and conductivity?

A2. Authors thank the reviewer for this comment. The porosity (f) of the fabricated composites is calculated from the density of macro-porous GCR composites (ρf) and the density of solid rGO/CNT/NR composites (ρs) using the following Equation: (Sensors, 2019, 19, 1968)

The porosity of GCR composites is shown in Figure 4b. G1.5C1.5R composites display the highest porosity (65.8%), while G0.5C0.5R composites possess the lowest one (41.2%), (See the Line 224-228, Page 6). Authors agree with reviewer that the degree of foaming can have an effect on the performance, but as shown in Figure 4c and 4d it is not the only parameter to affect the final properties of the composites. However, to take into account the reviewer comment, the specific electrical conductivity was calculated and shown in the revised version of the paper (Figure 4d).

Q3. Can any other material be used instead of NR Latex? If yes, what are the alternatives and why?

A3. Authors thank the reviewer for this comment. The NR latex can be substituted by synthetic rubber latex, such as styrene butadiene latex, butyl latex, and so on. In fact, the main reason for the formation of rGO/CNT/NR hydrogel is ascribed to the rGO and CNTs coated NR particles which, firmly interconnecting to each other through strong interactions, preserve the porous structure against the forces generated by the nitrogen gas evolving during chemical reduction. In another words, the entanglement effect of flexible rubber chains at the interface between two rubber particles results in the formation of hydrogels, and the macro-porous composites are consequently obtained after drying in the oven. Therefore, the method referred in this manuscript can be also applied to prepared macro-porous synthetic rubber composites with segregated network. In the section of results and discussion, the mechanism of the formation for rGO/CNT/NR hydrogels is described (Line 176-180, Page 4). In the section of conclusion, we added the following sentence to highlight the universality of this method: “The method referred in this manuscript can open a window in preparing the rubber composites with macro-porous structure and segregated network.

Reviewer 2 Report

The authors prepared polymer nanocomposite with rubber latex and GO and CNT to study the Joule heating behavior of their samples. I express my strong doubt on the novelty and more importantly application of their proposed materials in this field. They are also major improvements that should be done for clarification of their work and their results. I give the authors another opportunity to extensively revise their work and resubmit a revision again.

Author Response

Reviewer 2:

Q1. What is the rationale behind sample contents? For it doesn’t follow any pattern to extract any meaning full results. For instance, there three samples with G/C ratio equals to 1.

A1. Thank you for this comments. The aim of this manuscript is to investigate the effect of both conductive filler content and rGO/CNT ratio on electric heating behavior of macro-porous GCR composites. Therefore, we prepared two composite series. The first one includes the composites with the same rGO/CNTs ratio and different total filler content, such as G0.5C0.5R, G1.5C1.5R, and G3C3R. The second series includes the composites with the same total filler content (3wt%) and different rGO/CNTs ratio, such as G0.5C2.5R, G1.5C1.5R, and G2.5C0.5R. This approach allows us to get useful info about the importance of filler content and rGO/CNT content ratio on the electric heating properties of porous composites.

Q2. Authors should explain how did they find suitable amount of N2H4 for reduction. How do they claim that the GO reduced in this sample preparation? Does N2H4 have any side impacts on rubber or CNTs?

A2. Thank you for this comments. In our previous paper (Macromol. Mater. Eng., 2011; 296, 590, J. Mater. Chem., 2012, 22, 10464, Chem. Eng. J., 2018, 344,184), we clearly described the procedure of rGO preparation from GO by using N2H4 as a reducing agent. Considering the space limit, we didn’t describe the reduced procedure of GO in this manuscript. In addition, N2H4 can be removed easily during the drying because of its good solubility and volatility. Several papers focused on NR use hydrazine as reducing agent of GO, and it is considered as not affecting the polymer, working mainly on GO.

Q3. How did authors claim they develop segregated structure? There is no evidence or any references they mentioned that combination of these fillers leads to the segregated structure. The formation of segregated structures also should be schematically presented.

A3. Thank you for this comments. We describe the formation mechanism of segregated structure in the text. During the reduction process of GO, CNTs and the generated rGO coated the surface of NR particles, and as result of the electrostatic and π-π interactions establishing between the latex particles, the water dispersion turns to the rGO/CNT/NR hydrogels. In this way, the fillers coated on the surface of NR particles give rise to the formation of the segregated network (Line 169-172, Page 4). Based on TEM images (Figure 3), the segregated network of rGO and CNTs were found in the bulk material. In addition, some reports for the formation of segregated network in polymer matrix were published by some of the authors and cited in this manuscript, such as, J. Mater. Chem., 2012, 22, 10464, Chem. Eng. J., 2018, 344,184, Nanoscale 2020, 12, 7782, Nanoscale 2019, 11, 1011 and so on.

Q4. Author should also provide detailed characterization of their synthesized graphene e.g. size, elemental analysis and Raman for their synthesized graphene.

A4. Thank you very much. In our previous paper (Macromol. Mater. Eng., 2011, 296, 590), we clearly described the characterization of their synthesized rGO, such as size, elemental analysis and Raman. For the convenience of readers, we have added this sentence in the revised manuscript (Line 161 Page 4):

The size of rGO as well as results from XRD, XPS, FTIR and Raman characterizations were detailed in our previous paper. [33]”.

Q5. The method author used for estimation of density is completely wrong owing to their porous samples. M and V measurements to calculate density is valid for solid nonporous samples.

A5. Thank you very much. In this paper, we research the effect of filler content on their apparent density. The density of foams was measured in accordance with ISO 845 standard. The method referred in this paper was used also to calculate the density of foams in some papers, such as Macromolecules 2016, 49, 14, 5059–5067, Cement and Concrete Composites 2018, 87, 164-171, Polymer, 2018, 149, 134-145. Therefore, authors think that the method can be used to calculate the density of polymer foams.

Q6. How is the solution stability of GO/CNT? Does it show tyndall effect? What will happen to samples if settled for 2 hours or 24 hours?

A6. Thank you for this comments. The stability of rGO/CNT solution is very important for fabricating the GCR composites with uniform segregated network. The photo images of GO/CNT suspension standing for different amounts of time were shown in Figure 1a. Thanks to the amphiphilic nature of GO sheets which possess polar groups, such as hydroxyl, carboxyl and epoxy groups, and serve as surfactant, the hydrophobic CNTs can be easily dispersed into water under ultrasound. Notably, the rGO/CNTs are relatively stable in water over a period of 24 h. Tyndall effect is a good method to claim the solution stability of filler. We also try to find Tyndall effect. Although the GO/CNT is uniformly dispersed into water, the Tyndall effect is not found because of the high filler content (3-8 mg/ml).

Figure 1. The photo images of GO/CNT suspension standing for different amounts of time (a), GO/CNT/NR latex (b) and GCR macro-porous composites (c)

Q7. Authors should add TGA data for all the fillers and polymer as well for made a meaning full comparison. Author should also explain why their materials have low thermal stability? Their samples almost completely decomposed @450°C.

A7. Thank you for your comment. Authors agree with the reviewer’s comments. In the revision of this manuscript, the TGA curves of CNTs, rGO, NR and macro-porous GCR composites in air atmosphere were evaluated. Based on the Figure 7b, the onset degradation temperature of pristine NR without macro-porous structure and macro-porous GCR composites are more than 250 oC and they completely decomposed at about 450°C. Therefore, the introduction of fillers and porous structure don’t affect the thermal stability of NR.

Figure 7 and the text added to the manuscript is reported hereafter:

Figure 7. Thermal degradation curves of macro-porous GCR composites at different fillers concentration in a nitrogen (a) and air (b) atmosphere. The inset image of Figure 7b is the TGA curves of CNTs, rGO and pure NR in the air atmosphere.

“The thermal stability of electric heating materials affects the temperature range wherein the samples can operate. Figure 7 shows the thermal degradation curve of macro-porous GCR composites with different fillers content in a nitrogen (a) and air (b) atmosphere. For the thermal degradation curves of GCR composites in the nitrogen atmosphere (Figure 7a), the onset degradation temperature of G0.5C0.5R composite is the lowest among all the samples. Macro-porous GCR composites with higher filler contents exhibit higher onset degradation temperature, ascribing to the presence of carbon based fillers that contribute to thermally protect the polymer and hinder the transport of degradation gases/volatiles. [48] The thermal stability of macro-porous GCR composites were measured in the air atmosphere in order to evaluate their safe use as electrical heating materials, under ambient conditions, and the results are shown in Figure 7b. The inset image of Figure 7b displays that CNTs possess the most excellent thermal stability as compared with rGO and NR without macro-porous structure. In the case of macro-porous GCR composites, their onset degradation temperature exceeds the 250 °C which is the onset for the pristine NR without macro-porous structure. Therefore, the introduction of fillers and porous structure contribute to increase the thermal stability of NR. Based on Figure 7, macro-porous GCR composites can be safely used below 200 °C, which represents the maximum temperature reached during the electrical heating experiments.”

Q8. How much weight loss does the samples experience applying voltage for long period of times (1 hr)? The application they propose is the potential replacement for metal filament. Does this polymer nanocomposite have any good performance after 1 hr and applied voltage of 20v?

A8. Thank you for this comments. Based on Figure 5b, the steady-state temperature of G3C3R composites reach to 92.6 oC. Noteworthy, the GCR composites are thermally stable and not decomposed when the temperature does not exceed 200 oC (Figure 7). The results claim that GCR composites can work for a long period of times at an applied voltage of 20V.

Reviewer 3 Report

  1. In line 125, Table 1, you did not show the content or mass for N2H4. Please attend.
  2. In line 248, stated "An input voltage of 15 V was applied to all macro-porous GCR composites for investigating the effect of the ratio of CNTs and rGO, and filler contents on their electric heating behavior." Nevertheless, in Figure 5c not show the behavior of composite G0.5C0.5R in the graph, and no discussion about it.
  3. In line 273, a mention about Figure 5b is miscorrect. Must be, Figure 5c.
  4. In line 285, Table 2, again not show the characteristic parameters for composite G0.5C0.5R in the table, and no discussion about it.
  5. In line 314 to 316, stated "Based on Figure 7, macro-porous GCR composites can be safely used below 200 °C, which represents the temperature onset for thermal degradation of samples as determined by TGA in nitrogen atmosphere." Thereby, must be more interesting to perform the same analysis in air atmosphere, to compare the behaviour of composites and suggest their safe use, under ambient conditions.

Author Response

Reviewer 3:

Q1. In line 125, Table 1, you did not show the content or mass for N2H4. Please attend.

A1. Thank you for your comment. The volume of N2H4 are added in the revised Table 1, which now reads as follow:

Table 1. Formulations of GCR marco-porous composites

Sample

GO(g)

rGO(g)

CNT(g)

H2O (g)

NRL(60%)

N2H4(ml)

G0.5C0.5R

0.25

0.15

0.15

100

50

0.25

G0.5C2.5R

0.08

0.05

0.25

100

16.67

0.08

G1.5C1.5R

0.25

0.15

0.15

100

16.67

0.25

G2.5C0.5R

0.42

0.25

0.05

100

16.67

0.42

G3C3R

0.5

0.30

0.30

200

16.67

0.50

Q2. In line 248, stated "An input voltage of 15 V was applied to all macro-porous GCR composites for investigating the effect of the ratio of CNTs and rGO, and filler contents on their electric heating behavior." Nevertheless, in Figure 5c not show the behavior of composite G0.5C0.5R in the graph, and no discussion about it.

A2. Thank you for your comment. G0.5C0.5R composites are not suitable for the application of electric heater, because of their low electrical conductivity (4.7×10-4 S/m, Figure 4b). Therefore, we did not measure the electric heating behavior of G0.5C0.5R composites.

Following is the text added in the manuscript (Line 265-269, Page 8):

“An input voltage of 15 V was applied to the macro-porous GCR composites with good electrical conductivity for investigating the effect of the ratio of CNTs and rGO, and filler contents on their electric heating behavior. The results are shown in Figure 5c. G0.5C0.5R composites were not tested as electric heater materials, because of their low electrical conductivity (4.7×10-4 S/m, Figure 4b).”

Q3. In line 273, a mention about Figure 5b is miscorrect. Must be, Figure 5c.

A3. Thank you for this comment. It is revised.

Q4. In line 285, Table 2, again not show the characteristic parameters for composite G0.5C0.5R in the table, and no discussion about it.

A4. Thank you for your comment. G0.5C0.5R composites are not suitable for the application of electric heater, because of their low electrical conductivity (4.7×10-4 S/m, Figure 4b). Therefore, we did not measure the electric heating behavior of G0.5C0.5R composites. Please consider answer A2.

Q5. In line 314 to 316, stated "Based on Figure 7, macro-porous GCR composites can be safely used below 200 °C, which represents the temperature onset for thermal degradation of samples as determined by TGA in nitrogen atmosphere." Thereby, must be more interesting to perform the same analysis in air atmosphere, to compare the behaviour of composites and suggest their safe use, under ambient conditions.

A5. Thank you for your comment. Authors agree with the reviewer’s comments that the thermal stability of macro-porous GCR composites in air atmosphere is more interesting to compare their behavior and safe use under ambient conditions. In the revision of this manuscript, the TGA curves of CNTs, rGO, NR and macro-porous GCR composites in the air atmosphere were evaluated.

Figure 7 and the text added to the manuscript is reported hereafter:

Figure 7. Thermal degradation curves of macro-porous GCR composites at different fillers concentration in a nitrogen (a) and air (b) atmosphere. The inset image of Figure 7b is the TGA curves of CNTs, rGO and pure NR in the air atmosphere.

“The thermal stability of electric heating materials affects the temperature range wherein the samples can operate. Figure 7 shows the thermal degradation curve of macro-porous GCR composites with different fillers content in a nitrogen (a) and air (b) atmosphere. For the thermal degradation curves of GCR composites in the nitrogen atmosphere (Figure 7a), the onset degradation temperature of G0.5C0.5R composite is the lowest among all the samples. Macro-porous GCR composites with higher filler contents exhibit higher onset degradation temperature, ascribing to the presence of carbon based fillers that contribute to thermally protect the polymer and hinder the transport of degradation gases/volatiles. [48] The thermal stability of macro-porous GCR composites were measured in the air atmosphere in order to evaluate their safe use as electrical heating materials, under ambient conditions, and the results are shown in Figure 7b. The inset image of Figure 7b displays that CNTs possess the most excellent thermal stability as compared with rGO and NR without macro-porous structure. In the case of macro-porous GCR composites, their onset degradation temperature exceeds the 250 °C which is the onset for the pristine NR without macro-porous structure. Therefore, the introduction of fillers and porous structure contribute to increase the thermal stability of NR. Based on Figure 7, macro-porous GCR composites can be safely used below 200 °C, which represents the maximum temperature reached during the electrical heating experiments.”

Round 2

Reviewer 2 Report

The authors provide enough information to waive my concerns. I think the paper is publishable in polymers journal in its current form.